# Dynamic Modulation of SO_2_ Atmosphere for Enhanced Fresh-Keeping of Grapes Using a Novel Starch-Based Biodegradable Foam Packaging

**DOI:** 10.3390/foods12112222

**Published:** 2023-05-31

**Authors:** Shihua Mai, Yue Ma, Hongsheng Liu, Chao Li, Yuqing Song, Kaizhen Hu, Xinyan Chen, Ying Chen, Wei Zou

**Affiliations:** 1School of Food Science and Engineering, South China University of Technology, Guangzhou 510641, China; femaish@mail.scut.edu.cn (S.M.); feyuem@mail.scut.edu.cn (Y.M.); liuhongsheng@scut.edu.cn (H.L.); fe201920124894@mail.scut.edu.cn (C.L.); 202121028199@mail.scut.edu.cn (Y.S.); mehukaizhen@mail.scut.edu.cn (K.H.); 202130310114@mail.scut.edu.cn (X.C.); 2Sino-Singapore International Joint Research Institute, Guangzhou 510663, China; 3School of Food Science and Engineering, Yangzhou University, Yangzhou 225127, China; 008265@yzu.edu.cn; 4Department of Food Science and Technology, National University of Singapore, Science Drive 2, Singapore 117542, Singapore

**Keywords:** starch-based foam package, sandwich-like inner structure, cushioning, antisepsis, fresh-keeping

## Abstract

To improve the fresh-keeping of highly perishable fruits with high commercial value, a novel starch-based foam packaging material was developed in this study. The foam incorporated the antiseptic ingredient Na_2_S_2_O_5_, which chemically interacted with environmental moisture to release SO_2_ as an antifungal agent. Scanning electron microscopy (SEM), moisture absorption and mechanical measurements were used to characterize the unique sandwich-like inner structure of the foam which allowed for the modulable release of SO_2_. The starch-based foam exhibited sufficient resilience (~100%) to provide ideal cushioning to prevent physical damage to fresh fruits during transportation. When 25 g/m^2^ of Na_2_S_2_O_5_ was applied, the foam stably released over 100 ppm SO_2_ and demonstrated satisfactory antifungal performance (inhibition over 60%) in terms of maintaining the appearance and nutritional values (such as soluble solids 14 vs. 11%, total acidity 0.45 vs. 0.30%, and Vitamin C 3.4 vs. 2.5 mg/100 g) of fresh grapes during a 21 day storage period. Additionally, the residual SO_2_ (14 mg/kg) also meets the safety limits (<30 mg/kg). These research findings suggest great potential for the utilization of this novel foam in the food industry.

## 1. Introduction

Foam is a commonly used packaging material for fruit preservation and protection. Foam boxes, typically made from polystyrene (PS), can provide partial protection against bruising caused by external physical damage. However, they are not effective in delaying the natural decay process caused by microbial metabolism. For example, grapes, being a typical type of berry fruit, are highly susceptible to the gray mold pathogen Botrytis cinerea Pers [1,2,3]. The growth of this pathogen depends on various factors such as temperature and humidity during storage [4].

To minimize the economic loss of post-harvest fruit, Na_2_S_2_O_5_, a fungicide, has been used to chemically react with moisture from the fruit surface or surroundings, leading to the release of sulfur dioxide (SO_2_) to effectively kill harmful microbes within the packaging box [5,6,7]. However, this approach has limitations: traditional packaging materials cannot achieve long-term release of fungicides and non-biodegradable petrochemical-based foam poses a significant burden when disposed into the environment.

Starch foam has recently gained attention as a substitute for petrochemical-based products due to its abundant and renewable nature, low cost, and biodegradability. It has been particularly used in loose-fill packaging applications [8]. Additionally, starch foam is known for its native hydrophilicity, which makes it favorable for moisture absorption and retention [9,10]. This property has traditionally been utilized in modified atmosphere packaging, with the expectation of achieving a long-term antifungal effect on susceptible berry fruits.

To achieve this goal, a novel strategy was proposed in this study, involving the development of a starch-based foam with a sandwich-like structure that incorporates pre-loaded Na_2_S_2_O_5_ powder between two starch foam sheets. It is expected that this structure will enable the foam to absorb moisture from fruit respiration or the external environment, and then chemically react with Na_2_S_2_O_5_ to release SO_2_ as an antifungal ingredient in a modulable manner. Previous research has indicated the potential of utilizing composite packaging materials to modify the environmental atmosphere and improve the preservation of fruits. For instance, the effectiveness of aloe vera composite packaging has been demonstrated in suppressing the growth of aerobic mesophilic bacteria, yeast, and mold. This ultimately helps to preserve the essential nutrients and minimize loss during the fruit’s shelf life [11]. Multiple techniques, including SEM observation, moisture absorption, and mechanical measurements, were employed to characterize the unique functional structure and release characteristics of SO_2_. The antimicrobial effects and real performance of this novel starch-based biodegradable foam on fresh grapes were also assessed. It is hypothesized that this innovative foam could serve as a long-term effective fresh-keeping shield for susceptible perishable fruits with high commercial value.

## 2. Materials and Methods

### 2.1. Materials and Reagents

In this work, the starch-based foam was prepared using hydroxypropyl cornstarch with about 25% amylose content, which was provided by Derui Starch Co., (Luohe, China) and the initial water content was about 13%. Polyvinyl alcohol (078-20) and glycerol were obtained from Sinopec Petrochemical Co., Ltd. (Shanghai, China). Calcium carbonate was purchased from Yifeng Co. (Guangzhou, China). Na_2_S_2_O_5_ was supplied by Aladdin Chemical Co., (Shanghai, China). All chemicals used were of analytical grade. The Botrytis cinerea was from BeNa Culture Collection Co., Ltd. (Xinyang, China), and Potato Dextrose Agar (PDA) medium was from BeNa Culture Collection Co., Ltd. (Xinyang, China).

‘Kyoho’ grapes were purchased from local fruit market (Guangzhou, China). The preservation experiment was performed in 5.5 L airtight containers at room temperature (about 25 °C).

### 2.2. Starch-Based Foam Preparation

Starch-based foam was produced by two-step extrusions according to our previous method [12]. Firstly, hydroxypropyl cornstarch (80% *w*/*w*), glycerol (5% *w*/*w*), polyvinyl alcohol (PVA) (5% *w*/*w*), fiber (4% *w*/*w*), and water (6% *w*/*w*) were homogeneously mixed by a high-speed mixer (Shende SRZ-500, Zhangjiagang, China), and then were processed into pellets using a twin-screw co-rotating extruder (Keya, Nanjing, China) at a speed of 120 rpm. The temperature profile of the barrels was set at 30, 30, 80, 110, 110, 110, 80, and 80 °C, respectively. Subsequently, the pellets were dried for a constant moisture content of 16% and fed into a single-screw extruder (Tongjia, Jining, China) at a speed of 180 rpm. The temperature profile of the barrels was of 100, 180, 180 °C, and a sheet die with 400 mm width was heated to 200 °C. After it, the foams were cut into sheets with a size of 4 cm × 3 cm.

In this study, the starch-based foam with dynamic modulation of SO_2_ atmosphere was newly designed and the preparation procedure was shown in Figure 1. Initially, Na_2_S_2_O_5_ was evenly spread and loaded between two starch foam sheets, which were subsequently pressed together by using a hot-pressing method. In this case, a little amount of water was used as an adhesive and sprinkled on the narrow inner edge of the sheets, improving starch foam sticking without interference with the chemical reaction. In order to ensure that the residual concentration of SO_2_ remains below the safe limit of 30 mg/kg, the concentration range of Na_2_S_2_O_5_ was carefully selected through trial testing to meet the necessary safety requirements. The objective of the study is to achieve a Na_2_S_2_O_5_ concentration that is as low as possible while still providing effective antiseptic performance against microbial decay in grapes. This approach will also help reduce the manufacturing cost of the novel starch-based foam packaging. Thus, different dosages of Na_2_S_2_O_5_ (5, 15 and 25 g/m^2^) were used. The control group was also prepared without the addition of Na_2_S_2_O_5_. In the study, Na_2_S_2_O_5_ was manually cast into the starch-based pads. However, it is important to note that for practical applications, the process should be automated using a machine to ensure the most even distribution possible.

### 2.3. Scanning Electron Microscope (SEM)

SEM (EVO18, Zeiss, Oberkochen, Germany) was used to observe the raw materials and the cross-section of starch-based foams. A lower voltage of 10 kV was used during the test in order not to damage the cell structures. The samples were coated with gold before measurement [13].

### 2.4. Moisture Absorption

The objective of the starch-based foam package is to preserve and transport the locally grown “Kyoho grapes” in Guangzhou, China. These fresh grapes hold significant commercial value, but they typically ripen off the vine in August when the local humidity reaches high levels, often around 90–100% according to climate records released by NASA (Appendix A). Due to the grape granules’ vulnerability to rupture under humidity exceeding 95% [14], the main focus of the study is to investigate the efficacy of the starch-based foam package in effectively preserving the local “Kyoho grapes” within such extremely humid conditions. Foam sheets were first dried in an oven at 40 °C for 12 h and kept in the controlled containers with relative humidity (RH) of 86%, 93% and 98%, which were adjusted with saturated KCl, Na_2_SO_4_ and K_2_SO_4_ solutions, respectively. Absorbed moisture was calculated by the weight difference [15,16]. The absorption kinetics was investigated with a mathematical model proposed by Peleg (1988) [17], which was calculated using Equation (1):(1)$$M_{\left(t\right)}=M_{0}+\left(\frac{t}{K_{1}+K_{2}t}\right),$$
where *M*_(*t*)_ is the moisture at different time, *M*_0_ is the initial moisture, *t* is the incubation time, *K*_1_ is the Peleg flux constant, while *K*_2_ is Peleg capacity constant [10].

### 2.5. Compressive Strength, Resilience, and Recovery

The mechanical properties of the starch-based foams were measured by a universal testing machine (Instron, Model 5566, Norwood, MA, USA) after storage at the containers with different levels of RH for 1 week. The samples were fastened and compressed by a metal plate at a rate of 5 mm/min to a depth of 50% of its height. The detailed measurement was performed according to our previous study [12].

### 2.6. Artificial Vibrating Test

The packages were designed with and without starch-based foams, respectively, and then they were applied for grape preservation. Grapes were preserved by a one-day vibration testbed (M/MN100, Mooney Laboratory Equipment Co., Ltd., Dongguan, China) at a frequency of 2 Hz to simulate vibrating during transportation [18].

### 2.7. Measuring SO_2_ Content

The yield of SO_2_ was measured using a handheld gas analyzer [19] (Shenzhen Fukaite Technology Co., Ltd., Shenzhen, China). In the measurement, the foams loaded with Na_2_S_2_O_5_ were kept in an airtight container with a hole of 0.5 mm in diameter. Humidity of the container was adjusted according to the experimental requirement. The release of SO_2_ under room humidity of 86, 93, and 98% were recorded and calculated according to the growing indication of h, respectively. The indication was separated by 24 h.

### 2.8. Antimicrobial Analysis

Antimicrobial activity was determined experimentally according to a previous method [4,20]. The PDA medium plates were inoculated with Botrytis cinerea in different containers. Medium plates were used 3 times as replicates and the whole trial was repeated twice. The plates were incubated at 25 °C and radial fungal growth (mm) of Botrytis cinerea was measured after 7 days of incubation. The percentage of a reduction in colony diameter (CD) was calculated according to Equation (2):(2)$$\mathrm{CD}\left(\%\right)=\left(\frac{dc-dt}{dc}\right)\times 100,$$
where *dc* and *dt* are the average diameter of fungal growth in the control and treated samples, respectively.

### 2.9. Measuring Total Soluble Solids (TSS)

TSS value affects the taste as it indicates the level of sweetness of the fruit [5]. For the measurement of TSS, berries were randomly taken from the same bunch of grapes and squeezed into juice [21,22], and then measured with a digital hand-held refractometer (Shanghai Xinrui Electric Co., Ltd., Shanghai, China). Each replicate consisted of 3 grapes and every grape was measured three times.

### 2.10. Measuring Titratable Acid (TA)

TA measures the total acid concentration present in food items and in this work was determined according to the reference [23]. Grapes (10.0 g) were ground in a mortar and transferred into a 100 mL volumetric flask, kept for 30 min, and then were filtered. After it, 20 mL supernatants were transferred into a conical flask and 0.1 mL 1% (m/m) phenolphthalein solution was added, and then titrated by 0.01 mol/L sodium hydroxide (NaOH). The used NaOH solution was recorded to calculate TA. Each titration was repeated three times.

### 2.11. Measuring Vitamin C

Vitamin C was measured by the 2,6-dichloroindophenoltitration titration method as described in the previous references [24,25]. Grapes (10 g) were put into a mortar, and 50 mL of 20 g/L of the oxalic acid solution was poured and instantly mixed into a homogeneous solution. The homogenate was then fixed the volume to 100 mL with 20 g/L of the oxalic acid solution and shaken well, filtered 3 times. The trial was carried out in triplicate and the level of VC was calculated in 100 mg/g of fresh sample.

### 2.12. Measuring SO_2_ Residues

The residual SO_2_ was quantitatively measured by a distillation method according to the standard of GB 5009.34-2016 (China). Briefly, 5.0 g grapes were accurately weighed and SO_2_ is extracted by distillation in the iodine measuring bottle. Then titrating with iodine standard solution and recording the volume. The SO_2_ residues were calculated according to Equation (3)
(3)$$X(g/\mathrm{kg})=\frac{\left(V-V_{0}\right)\times 0.032\times c\times 1000}{m},$$
where *V* and *V*_0_ are the volume of iodine standard solution consumed for titration of grape samples and the blank group, respectively, *c* is the concentration of iodine standard solution, and m is the mass of grape samples.

### 2.13. Statistical Analysis

The experiment was arranged based on a completely randomized design with three replicates. One-way ANOVA was used to analyze the differences, and results with *p* < 0.05 were considered as significantly different.

## 3. Results

### 3.1. Starch-Based Foam’s Morphological Inner Structure

Figure 2 shows the ideal appearance of the starch-based foam sheet as a packaging material, with a smooth surface and regular wave-shape. In comparison, the cross-section of the foam sheet shows a high degree of porous inner cells under SEM [10,13], indicating excellent expansion necessary for ensuring adequate buffering and moisture absorption [13,26]. Due to the hydrophilic starch substrate, the interconnecting air cells gradually shrank or collapsed as the surrounding humidity increased from 60–98%, as shown in Figure 3.

### 3.2. Starch-Based Foam’s Moisture Absorption Behavior

To better understand the moisture absorption properties of the starch-based foam, it was subjected to room humidity of 86, 93, and 98%, as shown in Figure 4. The absorption process can be divided into two phases: an initial fast phase (from ~7 to 15% *w*/*w*) within the first three to five days, followed by a slower phase that gradually reached equilibrium by the end of the 21-day assay period. A higher room humidity (98 vs. 86%) resulted in faster absorption rates and higher maximal moisture absorption content (25 vs. 15% *w*/*w*), as confirmed by the Peleg kinetic model characterization of higher K_1_ and lower K_2_ values (Table 1). These findings suggest that water absorption occurs at a faster rate and larger capacity under higher environmental humidity, consistent with previous research [15,16,27].

### 3.3. Starch-Based Foam’s Mechanical Properties

The mechanical properties of starch foam, such as compressive strength and recovery ratio, depend on its inner loose-fill structure, which can vary under different environmental humidity conditions. Table 2 shows that the compressive strength of the foam was high at around 350–360 kPa under 60% humidity but dropped to only 80 kPa with an increase of humidity to 86%. With further increases in humidity, the compressive strength of the foam continued to decline, reaching only 15 kPa under 98% humidity. Such significant weakening of mechanical performance is disadvantageous for packaging materials and can be attributed to the plasticization effect caused by the absorption of more water molecules into the starch-based foam framework at higher environmental humidity.

In comparison, the Resilience Percentage of the starch-based foam remained almost unaltered (at ~105%) regardless of the increase of humidity from 60 to 98%. Additionally, the Recovery Percentage showed only a slight decrease from ~90 to 70%. This suggests that the starch-based foam can still play an ideal cushioning role as a fresh-package material, particularly under high humid conditions. The compression test is conducted to simulate the physical conditions experienced by the starch-based foam package during transportation. Its purpose is to verify whether the foam can withstand the considerable external pressure exerted on the package during fruit transportation. This test aims to ensure the reliability of the foam in protecting the fruits from physical damage prior to the onset of microbial decay.

All these mechanical properties can help maximally reduce the physical damage to the susceptible fruits during transportation. In the study, fresh grapes were packed using starch-based foam and subjected to a one-day assessment at 25 °C under artificial vibrating conditions without the application of the Na_2_S_2_O_5_ as a preservative (Figure 5). The control group exhibited several fallen granules, evident bruising, and fermentation smells (see a_1_ and a_2_). In contrast, the grapes packed inside the starch-based foam remained almost fresh without any visible physical damage (see b_1_ and b_2_). The starch-based foam was thus confirmed to play a satisfactory cushioning role in protecting susceptible fresh fruits during transportation [18,28,29].

### 3.4. Starch-Based Foam Preloaded with Na_2_S_2_O_5_ to Prevent Microbial Decay

To better preserve fresh fruits and prevent microbial decay, Na_2_S_2_O_5_ powder (interacting with environmental moisture for releasing the antiseptic SO_2_ atmosphere) is often applied in food preservation [30]. However, adding Na_2_S_2_O_5_ powder without any packaging is not an appropriate strategy, as it can result in the excessive and uncontrollable release of SO_2_ (red lines in Figure 6a–c). For instance, SO_2_ content released from Na_2_S_2_O_5_ (5 g/m^2^) under typical high humidity (98%) was initially high within the first three days but quickly declined afterwards (Figure 6c). In comparison, when pre-loaded into the hydrophilic starch substrate of the foam package with its unique sandwich-like inner structure, Na_2_S_2_O_5_ powder can release a more controlled amount of SO_2_ content in two phases during the storage assay. This is due to moisture being absorbed into the starch substrate, enabling a more controlled release of SO_2_.

The release of SO_2_ was within the range of antiseptic and food safety requirements, and pre-loading Na_2_S_2_O_5_ powder at 5 g/m^2^ was sufficient to generate enough safe content of SO_2_ atmosphere. Higher content (15 or 25 g/m^2^) resulted in excessive accumulation of SO_2_ atmosphere, which may incur safety risks (over 30 mg/kg *Codex Alimentarius International Food Standards*). Figure 7 aims to illustrate the starch-based foam package can release relatively low but stable concentration of SO_2_ during the 21-day trail. The safe level of SO_2_ often refer to its residual content on grapes, which is clearly indicated as 30 mg/kg in last Figure and well discussed below in Section 3.5.

The release of SO_2_ must comply with both antiseptic and food safety requirements. As shown in Figure 7, using 5 g/m^2^ of pre-loading Na_2_S_2_O_5_ powder may not be sufficient to generate a safe level of SO_2_ atmosphere compared to higher contents (15 or 25 g/m^2^) over a 21 day storage period, which may result in inadequate protection against microbial decay. On the other hand, excessive usage of Na_2_S_2_O_5_ poses a food safety risk as it may leave residual SO_2_ content on the fresh fruits above the threshold limit [31]. Both of these important aspects will be addressed in detail in the following sections.

A seven-day assay has been widely recognized in previous studies as an effective method to evaluate the efficacy of inhibiting microbial growth [4,20,32]. In this study, the same assay duration was employed to investigate whether the Na_2_S_2_O_5_ preloaded starch-based foam could effectively inhibit the growth of Botrytis cinerea, the main culprit behind the microbial decay of fresh grapes. During a seven-day assay, the Na_2_S_2_O_5_ preloaded starch-based foam was tested for effectiveness against the typical Botrytis cinerea (the main cause of fresh grapes’ microbial decay). The antimicrobial effectiveness was evaluated through the quantification of inhibition percentages (Figure 8a,b) and illustrated in the study [32]. The results showed that the antimicrobial performance was highly dependent on the concentration of antiseptic chemicals, with an inhibition percentage of above 50% observed when the Na_2_S_2_O_5_ concentration was over 25 g/m^2^. Additionally, the study found that the release of SO_2_ from Na_2_S_2_O_5_ was facilitated by the presence of more moisture within the enclosure. This was evidenced by a significant increase in the antimicrobial inhibition percentage when the humidity level was set to 93% compared to 86%. However, the excellent antimicrobial performance decreased when the humidity level exceeded 98%. Because this humid condition can promote the growth of various microbial communities, which may compromise the effectiveness of the preservation method.

### 3.5. The Fresh-Keeping Performance of Starch-Based Foam Preloaded with Na_2_S_2_O_5_

The use of antiseptic Na_2_S_2_O_5_ powder preloaded into starch-based foam has been shown to effectively preserve fresh grapes susceptible to Botrytis cinerea (Figure 9) [7,33,34]. The starch-based foam alone resulted in rapid visible decay of grapes from the sixth day of storage, but even a small addition of Na_2_S_2_O_5_ (just 5 g/m^2^) helped maintain the grapes’ fresh appearance for up to 18 days. Further increasing the amount of Na_2_S_2_O_5_ to 15–25 g/m^2^ preserved the fresh appearance of the grapes throughout the entire 21 day storage period. Total soluble solids, acidity, and vitamin C are recognized as the three key nutritional parameters of grapes, which tend to decrease due to microbial metabolism during storage [35]. The release of SO_2_ atmosphere from Na_2_S_2_O_5_ has been previously validated as an effective method to inhibit microbial growth and preserve the nutritional quality of grapes during storage [36] (Figure 10a–c). Because an appropriate content of Na_2_S_2_O_5_ (from 5 to 25 g/m^2^) was found to better preserve the total soluble solids, acidity, and Vitamin C, which would otherwise be quickly lost during storage. However, considering the common safety threshold of SO_2_ residues (30 mg/kg, U.S. Federal Registry, 1986), the results indicate that adding Na_2_S_2_O_5_ powder at ~25 g/m^2^ is ideal (only ~14 mg/kg of SO_2_ residues) for achieving a good balance between fresh-keeping effectiveness and food safety.

All these results indicate a significant improvement in the fresh-keeping ability of the starch-based foam packaging when the appropriate amount of antiseptic Na_2_S_2_O_5_ is included.

## 4. Conclusions

A novel starch-based foam was successfully developed in this study as a packaging material for fresh fruits, featuring a unique sandwich-like inner structure. This foam not only provided sufficient resilience to protect the fruits from physical damage during transportation but also facilitated the pre-loaded Na_2_S_2_O_5_ powder to chemically interact with environmental moisture, allowing for modulable release of the antiseptic SO_2_ atmosphere. As a result, the starch-based foam effectively preserved the appearance and key nutrients (such as soluble solids, acidity, and Vitamin C) of fresh grapes during a 21-day storage period, indicating its great potential for fresh-keeping of perishable fruits with high commercial value in the food industry.

In summary, the research successfully validates the hypothesis by developing a novel starch-based foam packaging that effectively preserves highly perishable fresh grapes of significant commercial value. However, there are several limitations that should be addressed in future studies. Firstly, there is a need for characterizing the grapes’ microbial stability and sensory properties during storage in the starch-based foam package. Furthermore, the applicability of the starch-based foam packaging in dry regions with lower humidity needs to be explored. Additionally, its potential for wider application to other perishable fruits beyond grapes should be investigated. Further applied research is necessary to optimize the dosage and distribution of Na_2_S_2_O_5_ to meet specific requirements and enhance the overall performance of the packaging system.

## Figures and Tables

**Figure 1 foods-12-02222-f001:**
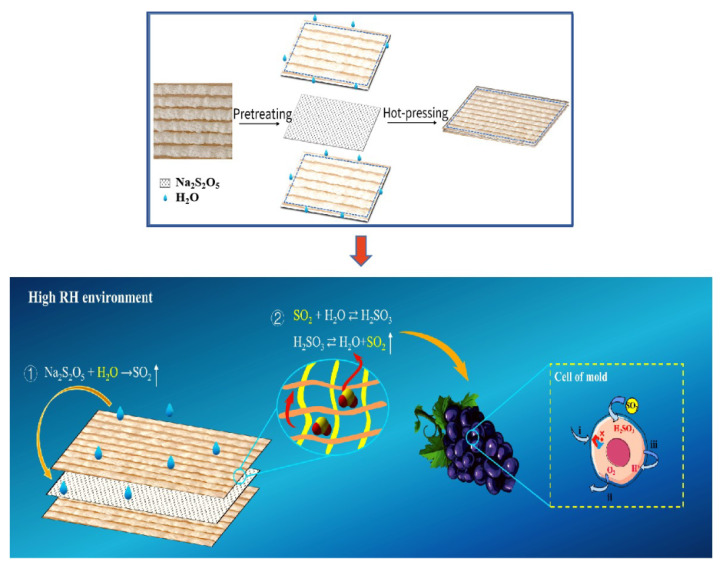
An illustration of starch-based foams pre-loaded with Na_2_S_2_O_5_ and its functions.

**Figure 2 foods-12-02222-f002:**
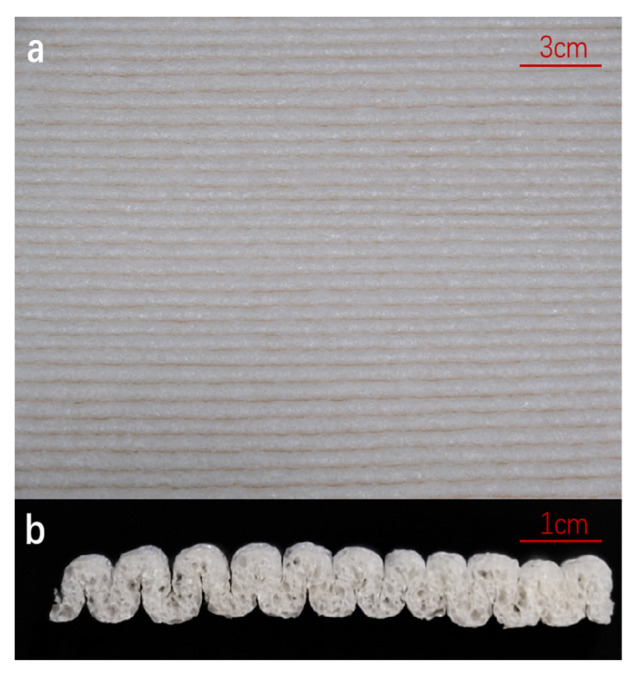
Photograph of surface (**a**) and cross-section (**b**) of starch-based foam.

**Figure 3 foods-12-02222-f003:**
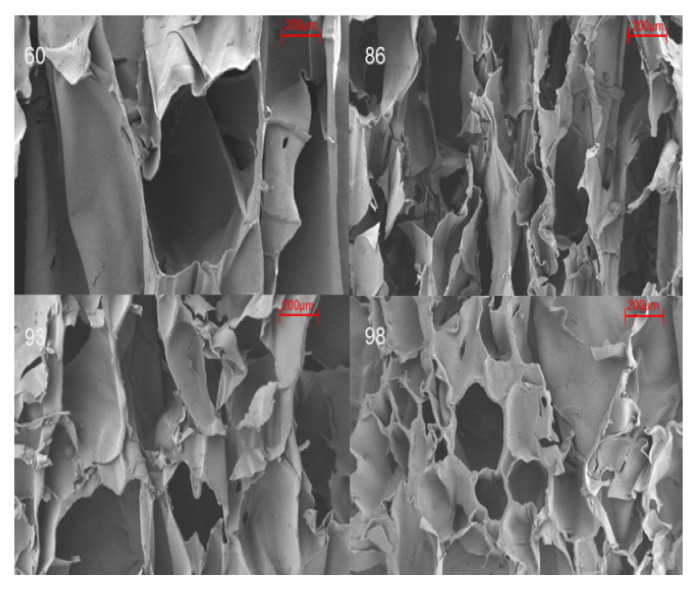
Inner-cell structure of starch-based foam under humidity levels of 60%, 86%, 93%, and 98%.

**Figure 4 foods-12-02222-f004:**
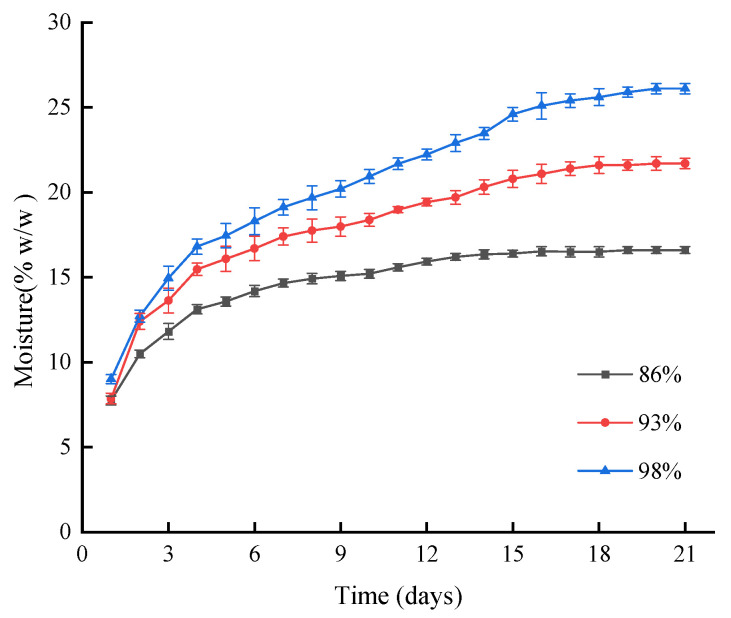
Moisture percentage of starch-based foam over time at humidity levels of 60%, 86%, 93%, and 98%.

**Figure 5 foods-12-02222-f005:**
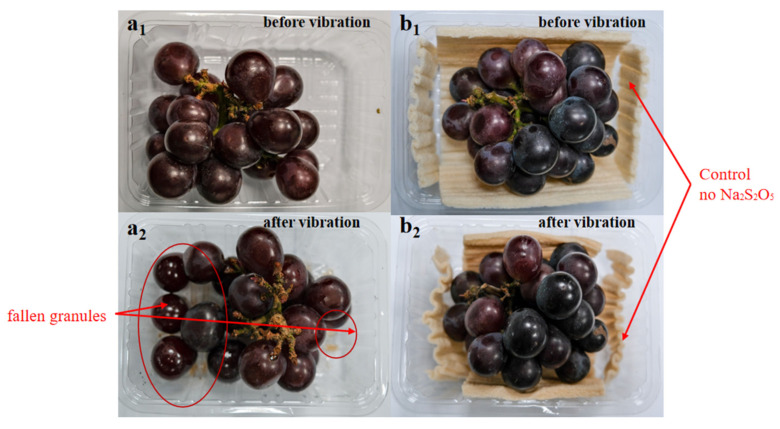
Fresh grapes with and without the starch-based foam package (**a_1_**,**b_1_**), and their respective appearance after one-day artificial vibrating assessment (**a_2_**,**b_2_**). No Na_2_S_2_O_5_ as preservative was applied in the test.

**Figure 6 foods-12-02222-f006:**
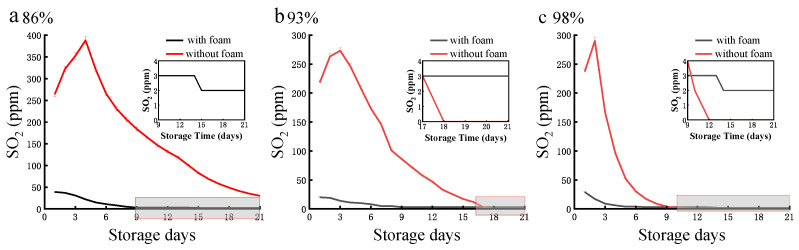
Time-dependent SO_2_ concentration released from Na_2_S_2_O_5_ (5 g/m^2^) with and without starch-based package foam at humidity levels of 86% (**a**), 93% (**b**), and 98% (**c**) respectively.

**Figure 7 foods-12-02222-f007:**
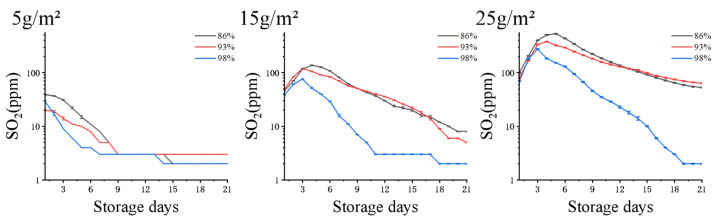
Time-dependent SO_2_ concentration released from starch-based foam package with varying Na_2_S_2_O_5_ amounts (5, 15, and 25 g/m^2^) at humidity levels of 86%, 93%, and 98%.

**Figure 8 foods-12-02222-f008:**
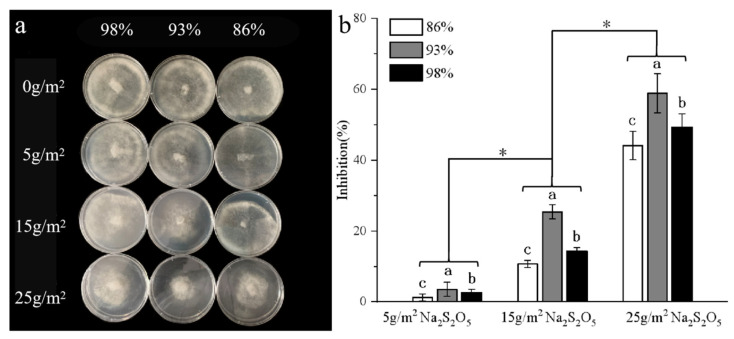
Antimicrobial activity (**a**) and the inhibition rate (**b**) of the starch-based foam with 0, 5, 15, and 25 g/m^2^ of Na_2_S_2_O_5_ at humidity levels of 86%, 93%, and 98%. One-way ANOVA method used; difference significant (in terms of a, b, c and *) at the 0.05 level.

**Figure 9 foods-12-02222-f009:**
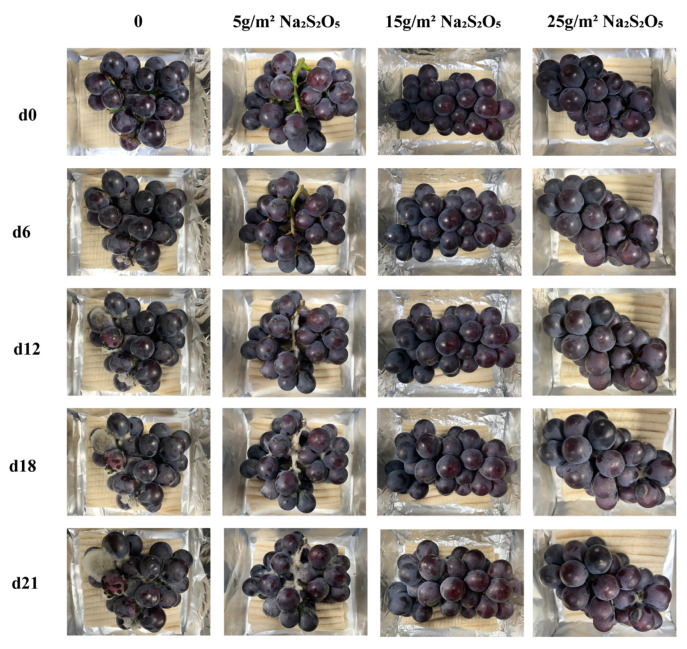
Appearance of fresh grapes packed with starch-based foam with varying Na_2_S_2_O_5_ amounts (5, 15, and 25 g/m^2^) over a 21-day period.

**Figure 10 foods-12-02222-f010:**
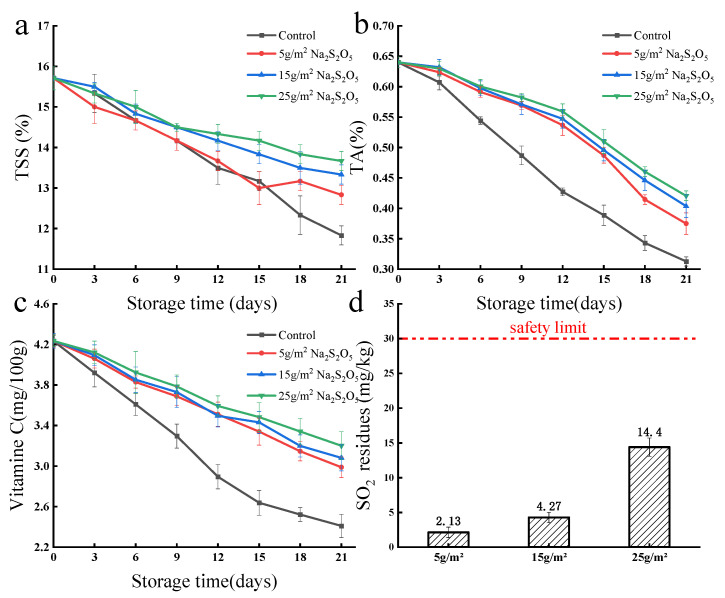
Changes in total soluble solids (TSS, (**a**)), acidity (TA, (**b**)), Vitamin C content (**c**), and SO_2_ residual concentration (**d**) of fresh grapes packed with starch-based foam containing varying amounts of Na_2_S_2_O_5_ (5, 15, and 25 g/m^2^) over a 21-day period.

**Table 1 foods-12-02222-t001:** Compressive properties of foam sheets at different humidity levels.

Humidity Levels	First Compressive Strength (Kpa)	Second Compressive Strength (Kpa)	Resilience (%)	Recovery (%)
Control (60%)	345.09 ± 3.46 ^a^	359.45 ± 7.40 ^a^	104.15 ± 1.10 ^b^	90.21 ± 1.43 ^a^
86%	80.08 ± 4.85 ^b^	83.73 ± 4.26 ^b^	104.63 ± 1.46 ^b^	84.72 ± 1.37 ^b^
93%	28.42 ± 1.34 ^c^	31.11 ± 1.23 ^c^	109.51 ± 1.50 ^a^	78.36 ± 2.06 ^c^
98%	15.09 ± 0.62 ^d^	15.97 ± 0.57 ^d^	105.82 ± 0.96 ^b^	70.19 ± 2.48 ^d^

One-way ANOVA method used; difference significant (in terms of a, b, c and d) at the 0.05 level.

**Table 2 foods-12-02222-t002:** Peleg model parameters of starch foams at humidity levels of 86%, 93%, and 98%.

Humidity Levels	K_1_	K_2_
86%	64.17	1.88
93%	74.29	1.42
98%	85.50	1.15

## Data Availability

Data is contained within the article (or Appendix A).

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
