# Peer review of "Dynamic Modulation of SO2 Atmosphere for Enhanced Fresh-Keeping of Grapes Using a Novel Starch-Based Biodegradable Foam Packaging"

_foods, 2023, doi:10.3390/foods12112222_

Round 1

Reviewer 1 Report

General comments

 This manuscript describes the development of a novel starch-based biodegradable fresh-keeping foam with dynamic modulation of SO2 atmosphere and evaluates its effectiveness to extend the shelf life of grapes. The study could be of interest to the field of fresh fruit preservation, and it was adequately designed and performed. However some clarifications and corrections are necessary.

 Specific comments

 Materials and methods

 Line 71-72 Rewrite the sentence to avoid the mixing between source of media and the culture collection for Botrytis cinerea.

 Labels in Figure 1 must be increased since it is difficult to read

 How was Na2S2O5 spread on starch?

 How were the dosages of  Na2S2O5 selected?

 What process is intended to model with the conditions selected in the compression test?

  Previously, I suggested including results about microbial stability and sensory characteristics of the product.

 Lines 133, 148 and 156, do not begin the sentence with a number

 Results

 Lines 191, 264 and 282: a space is missing between Figure and number

 Table 2 appears first to its mention

 Line 226, add the info that the starch foam did not contain the preservative

 Figure 5: grapes seems to be in the same type of container and damage is not clearly observed

 Figure 7: make and adequate identification of each panel and add a line indicating the safe level of SO2

 Line 264: explain why antimicrobial effectiveness was evaluated at 7th day

 Lines 285-290: Explain why quality parameters are preserved by SO2

quality of english is adequate

Author Response

All the reviewer's comments are well addressed accordingly. A word file is attached to include the revised details in terms of text, figures, and Tables. 

Thanks for your professional comments on our research article. 

Reviewer 2 Report

The manuscript with title: "A novel starch-based biodegradable fresh-keeping foam with dynamic modulation of SO2 atmosphere" is about modified packaging to improve the shelf life of grapes. In general the research is interesting. The objectives are clear and research design aligned with the objectives.

The manuscript is related to the journal's aims and scopes, and need improvement before final decision by the editor:

1- Title: Make it more interesting; try to focus on the Modified atmosphere packaging, and it will be more interesting for the readers.

2- Abstract: Support the results by some quantitative data.

3- Keywords: Choose a keyword about "Packaging".

4- Introduction: It needs to extend in terms of literature review. Try to expand the problem statement and improve the novelty statement. You can refer to this interesting article in the same field: Esmaeili, Yasaman, et al. "The synergistic effects of aloe vera gel and modified atmosphere packaging on the quality of strawberry fruit." Journal of Food Processing and Preservation 45.12 (2021): e16003.

5- Methods: All methods should have a proper reference(s) for example check 2.7: Measuring SO2 content,...

6- Results: Complete the statistical analysis by adding results of inferential statistics in Table 2 and Figure 8.

7- Conclusion: Focus on hypothesis justification and future research recommendations.

Author Response

(The authors gave the same response as above.)

Reviewer 3 Report

The manuscript presents interesting experiment of biodegradable active packaging. The structure of the manuscript is correct. The results are properly and clearly discussed. My main concern is the range of RH tested, what affect almost all of the presented data. Why authors decided to study such high values?

Minor remarks:

Line 9-10 please double check, the initial part is repeated

Line 79 what were the proportions of ingredients?

Line 94 the dosage is not clear it should be g per selected unit

Figure 1 please enhance the size and resolution, currently they are not readable

Line 127 please state the humidity

Line 326 editorial

Author Response

(The authors gave the same response as above.)

Round 2

Reviewer 3 Report

The manuscript has been improved. Thank you for corrections.